# Insights from a workplace SARS-CoV-2 specimen collection program, with genomes placed into global sequence phylogeny

Owen P. Leiser[1], Deanna L. Auberry[2], Erica Bakker[2], Will Chrisler[3], Kristin Engbrecht[2], Heather Engelmann[2], Sarah Fansler[4], Vincent Gerbasi[3], Joshua Hansen[2], Chelsea Hutchinson[5], Janine Hutchison[2], Mary J. Lancaster[2], Kathleen Lawrence[3], Angela Melville[2], Jennifer Mobberley[2], Isabelle O'Bryon[2], Kristie L. Oxford[3], Tessa Oxford[2], Shelby Phillips[2], Kabrena E. Rodda[6], James A. Sanford[3], Athena Schepmoes[3], Brian E. Staley[7], Kelcey Terrell[7], Kristin Victry[2], Cynthia Warner[2], Kristin M. Omberg[2]*

**1** Pacific Northwest National Laboratory, Chemical and Biological Signatures, Seattle, WA, United States of America, **2** Pacific Northwest National Laboratory, Chemical and Biological Signatures, Richland, WA, United States of America, **3** Pacific Northwest National Laboratory, Biological Systems Science, Richland, WA, United States of America, **4** Pacific Northwest National Laboratory, Environmental Protection & Regulatory Programs, Richland, WA, United States of America, **5** Pacific Northwest National Laboratory, Integrative Omics, Richland, WA, United States of America, **6** Pacific Northwest National Laboratory, Radiochemical Analysis, Richland, WA, United States of America, **7** InCyte Diagnostics, Richland, WA, United States of America

* Kristin.Omberg@pnnl.gov

## Abstract

In 2020, the Department of Energy established the National Virtual Biotechnology Laboratory (NVBL) to address key challenges associated with COVID-19. As part of that effort, Pacific Northwest National Laboratory (PNNL) established a capability to collect and analyze specimens from employees who self-reported symptoms consistent with the disease. During the spring and fall of 2021, 688 specimens were screened for SARS-CoV-2, with 64 (9.3%) testing positive using reverse-transcriptase quantitative PCR (RT-qPCR). Of these, 36 samples were released for research. All 36 positive samples released for research were sequenced and genotyped. Here, the relationship between patient age and viral load as measured by Ct values was measured and determined to be only weakly significant. Consensus sequences for each sample were placed into a global phylogeny and transmission dynamics were investigated, revealing that the closest relative for many samples was from outside of Washington state, indicating mixing of viral pools within geographic regions.

## Introduction

SARS-COV-2 coronavirus has caused nearly 80 million cases of COVID-19 in the United States since the onset of the global pandemic in late 2019/early 2020, and approximately 975,000 deaths [1]. COVID-19 can lead to severe negative health outcomes in recovered patients, including neurological [2,3] and cardiovascular [4,5] effects, with an estimate that as many as 80% of recovered COVID-19 patients experience some form of long-term health

**Data Availability Statement:** Sequence reads produced during this study have been deposited in

the NCBI Sequence Read Archive under accession PRJNA891756.

**Funding:** This research was supported by the DOE Office of Science through the National Virtual Biotechnology Laboratory, a consortium of DOE national laboratories focused on response to COVID-19, with funding provided by the Coronavirus CARES Act. The funders had no role in study design, data collection and analysis, decision to publish, or preparation of the manuscript.

**Competing interests:** The authors have declared that no competing interests exist.

consequence including fatigue, headaches, attention disorder, hair loss, and dyspnea (difficulty breathing) [6]. Among those infected by COVID-19, advanced age is associated with increased disease severity and negative outcomes [7,8].

In 2020, the US Department of Energy established the National Virtual Biotechnology Laboratory (NVBL) in March 2020 to address key challenges associated with the COVID-19 crisis [9]. NVBL brought together the broad scientific and technical expertise and resources of DOE's 17 national laboratories to help tackle medical supply shortages, discover potential drugs to fight the virus, develop and validate COVID-19 testing methods, model disease spread and impact across the nation, and understand virus transport in buildings and the environment. As part of that effort, Pacific Northwest National Laboratory, located in Richland WA, established a capability to conduct on-campus collection and analysis of nasopharyngeal specimens from employees self-reporting symptoms of COVID-19. In Washington state, USA, approximately 1.45 million cases and 12,000 deaths have been reported as of March 2022 [10]. Between January and October of 2021, 36 specimens collected by PNNL were identified as SARS-COV-2 positive by two quantitative PCR assays (N1 and N2) and submitted to the University of Washington Medicine's Virology Laboratory (Seattle WA) for genotyping and sequencing. An additional assay for human RNAse P (Rp) was used as an internal control. Our initial goal was to investigate the relationship between patient age and viral load (as measured by RT-qPCR cycle threshold (Ct) values), as suggested in [11] and/or SARS-COV-2 lineage. Here, we report the results of these efforts and place the samples in a global phylogeny.

## Materials and methods

This work was approved by the PNNL institutional review board (IRB number 2020–12 / PNNL000279) and informed consent given by all patients. No minors were involved in this study. Sampling was conducted at the on-site occupational health clinic at PNNL. A form containing information about the study and requesting written consent for research use was provided to patients when they registered for voluntary COVID testing. Samples were collected and processed according to CDC guidelines [CDC 2019-Novel Coronavirus (20190CoV) Real-Time RT-PCR Diagnostic Panel; [12]]. Steps are briefly outlined below.

### Sample collection and preparation

Nasopharyngeal swabs were collected at the PNNL onsite clinic and stored at 2–8˚ C for up to 36 hours after receipt by the laboratory. Any samples requiring additional time for processing were stored at -70˚ C for no more than 72 hours. Samples were processed inside of a class II biosafety cabinet at Biosafety Level 2 (BSL-2) controls. Viral nucleic acids were extracted using a Qiagen QIAamp Viral RNA Mini Kit according to manufacturer's instructions. Of 688 samples collected between January and October 2021, 64 were positive for SARS-CoV-2 and 36 samples were released for research. All 36 positive samples released for research were sequenced and genotyped.

### Quantitative PCR and statistical analysis

Reverse transcriptase-quantitative qPCR assays were prepared inside of a PCR workstation according to manufacturers' instructions. Sample nucleic acids, including positive and negative controls, were added to TaqPath 1-Step RT-qPCR Master Mix (ThermoFisher) and 2019-nCoV CDC Probe and Primer Kit for SARS-COV-2 (Biosearch Technologies) in 96-well plates and sealed with MicroAmp optical adhesive film (Applied Biosystems). RT-qPCR was carried out on an Applied Biosystems 7500 Fast Dx Real-Time PCR instrument using the following cycles: UNG incubation 25˚ C for 2 minutes, RT incubation 50˚ C for 15 minutes,

enzyme activation 95˚C for 2 minutes, 45 amplification cycles 95˚C for 3 seconds/55˚C for 30 seconds.

All statistical analyses of Ct values (i.e., Pearson correlations and Wilcoxon ranked-sum tests) were performed using R v4.2.0 and packages ggsignif v0.6.3 and ggpubr v0.4.0. Tests of normality for age cohorts was performed using the Shapiro-Wilk test function of base R.

### Library preparation and sequencing

Viral nucleic acids were submitted to the University of Washington Medicine's Virology Laboratory for sequencing. Libraries were prepared using Swift BioSciences SARS-CoV-2 amplicon panel according to manufacturer's instructions, resulting in 345 amplicons. Libraries were sequenced using an Illumina NextSeq500 in paired-end mode.

### Sequence processing

Sequence reads have been deposited in the NCBI Sequence Read Archive (BioProject accession PRJNA891756). Raw read files were preprocessed by removing low quality reads, low quality base calls from ends of reads, Illumina sequencing adapters, and residual PhiX sequences using the bbduk function of BBTools v37.93 (https://sourceforge.net/projects/bbmap/). Preprocessed reads were aligned to SARS-COV-2 Wuhan-Hu-1 (accession MN908947; [13]) using bwa mem v0.7.12 [14]. Sequences corresponding to primers used during sequence library preparation were trimmed from alignments using primerclip v0.3.8 (https://github.com/swiftbiosciences/primerclip). After sorting and indexing of alignment files using samtools v1.14 [15], variation from reference was determined using GATK HaplotypeCaller v4.2.0.0 [16]. Indels were normalized and filtered within 5 bases of adjacent indels, and calls were filtered to a minimum mapping quality of 20 and minimum sequence depth of 10 on a per-base basis using bcftools v1.14 [15], and consensus sequence contigs were generated from filtered calls. Contigs smaller than 500 bases were removed, and assembly statistics were generated using Quast v5.0.2 [17]. Finally, for quality control purposes any reads not mapping to reference were assembled *de novo* using SPAdes v3.9.0 [18] using careful setting and kmer lengths of 21, 33, 55, 77, 99, and 127.

### Alignment and local phylogenetic tree generation

Consensus sequences were aligned in Molecular Evolutionary Genetics Analysis (MEGA) v10.2.6 [19,20] using the MUSCLE algorithm. Maximum-likelihood tree of samples was generated in MEGA using the Tamura-Nei model of substitutions and assuming uniform rates, with 500 bootstraps and Wuhan-Hu-1 as root.

### Placement in global phylogeny

Consensus sequences were submitted to the UCSC UShER portal (http://hgw1.soe.ucsc.edu/cgi-bin/hgPhyloPlace; [21]) for placement into a global phylogenetic context. The phylogenetic tree version used contained 8,790,585 total genomes from the GISAID [22,23], GenBank [24], COG-UK [25], and CNCB [26] databases. Global and subtrees were downloaded directly from UShER output.

## Results and discussion

### Statistical analysis

36 samples collected between January and October 2021 from patients aged 19–71 were determined to be positive by qPCR assays targeting viral N1 and N2 genes, with an additional Rp internal control (Table 1, Fig 1). Using these data, Pearson correlation between patient age and

**Table 1. Samples used in this study.**

| Sample ID | Sequence ID | Lineage | Collection Month | Patient Age | Patient Gender | Ct N1 | Ct N2 | Ct Rp |
|---|---|---|---|---|---|---|---|---|
| 584081 | V341233082 | B.1.232 | January | 53 | Male | 19 | 18 | 28 |
| 687573 | V341233116 | B.1.577 | January | 52 | Female | 27 | 28 | 27 |
| 927695 | V341233129 | B.1.36.31 | January | 27 | Male | 22 | 22 | 29 |
| 946853 | V341233145 | B.1.2 | January | 43 | Male | 28 | 29 | 28 |
| 519389 | V341233160 | B.1.429 | January | 36 | Female | 17 | 17 | 30 |
| 355175 | V341233169 | B.1.311 | January | 65 | Female | 18 | 18 | 27 |
| 324854 | V341233183 | B.1.2 | January | 49 | Male | 17 | 17 | 32 |
| 723147 | V341233231 | B.1.232 | January | 46 | Female | 26 | 27 | 32 |
| 886266 | V341233246 | B.1.2 | January | 52 | Male | 21 | 22 | 32 |
| 487638 | V341233275 | B.1.576 | January | 50 | Female | 22 | 22 | 30 |
| 841525 | V341233299 | B.1.596 | February | 60 | Male | 19 | 20 | 31 |
| 354648 | V341233321 | B.1.2 | February | 47 | Female | 21 | 22 | 31 |
| 230366 | V341233343 | B.1.429 | February | 45 | Male | 24 | 24 | 30 |
| 735397 | V341233359 | B.1.2 | February | 38 | Female | 32 | 31 | 28 |
| 838436 | V341233381 | B.1.241 | March | 37 | Female | 30 | 30 | 26 |
| 237510 | V349346041 | B.1.1.7 | April | 53 | Female | 18 | 19 | 31 |
| 316866 | V349346159 | Q.1 | April | 51 | Male | 22 | 22 | 29 |
| 558580 | V349346303 | B.1.1.7 | April | 29 | Male | 20 | 22 | 26 |
| 250656 | V349346065 | AY.26 | August | 44 | Female | 21 | 20 | 25 |
| 281225 | V349346087 | AY.120.1 | August | 61 | Male | 26 | 26 | 30 |
| 415215 | V349346212 | B.1.621 | August | 54 | Male | 20 | 20 | 26 |
| 603888 | V349346344 | AY.120.1 | August | 63 | Female | 20 | 20 | 30 |
| 656785 | V349346358 | AY.120.1 | August | 53 | Female | 19 | 19 | 29 |
| 797176 | V349346383 | B.1.621 | August | 63 | Male | 18 | 18 | 27 |
| 962861 | V349346412 | AY.44 | August | 62 | Male | 23 | 23 | 28 |
| 285808 | V349346128 | AY.25.1 | September | 58 | Female | 26 | 26 | 31 |
| 343312 | V349346187 | AY.25 | September | 51 | Male | 23 | 22 | 30 |
| 434984 | V349346227 | AY.44 | September | 35 | Female | 19 | 19 | 23 |
| 478320 | V349346282 | AY.44 | September | 71 | Female | 19 | 21 | 30 |
| 923965 | V349346396 | AY.113 | September | 19 | Male | 19 | 20 | 27 |
| 987729 | V349346451 | AY.25.1 | September | 58 | Female | 22 | 22 | 26 |
| 517711 | NA | NA | October | 61 | Female | 27 | 28 | 25 |
| 218375 | V349346026 | AY.100 | October | 31 | Female | 20 | 19 | 28 |
| 246770 | V349346050 | AY120.1 | October | 40 | Male | 16 | 16 | 30 |
| 321833 | V349346177 | AY.20 | October | 61 | Male | 16 | 16 | 29 |
| 450350 | V349346250 | AY.122 | October | 59 | Male | 20 | 19 | 30 |

cycle threshold (Ct; Fig 1) values for all three assays was investigated using linear regression. No significant correlation was found between age and Ct value (inset, Fig 1).

To assess the relationship more finely between age and Ct values, qPCR results were binned for each assay into decades by birth years: 1950–1959 (n = 4), 1960–1969 (n = 12), 1970–1979 (n = 11), 1980–1989 (n = 5), 1990–1999 (n = 3), and 2000–2009 (n = 1), and pairwise comparisons made between each bin using Wilcoxon ranked-sum test. In this analysis, only two pairs of decades had significantly different mean Ct values (Fig 2): N1 1950–1959 vs 1970–1979 (21 vs 23 Ct; p = 0.0028), and N1 1950–1959 vs 1970–1979 (20 vs 23 Ct; p = 0.015. Together, these data suggest that while age is associated with more severe COVID-19 symptoms and morbidity [7,8], it does not necessarily correlate with viral load in all patients.

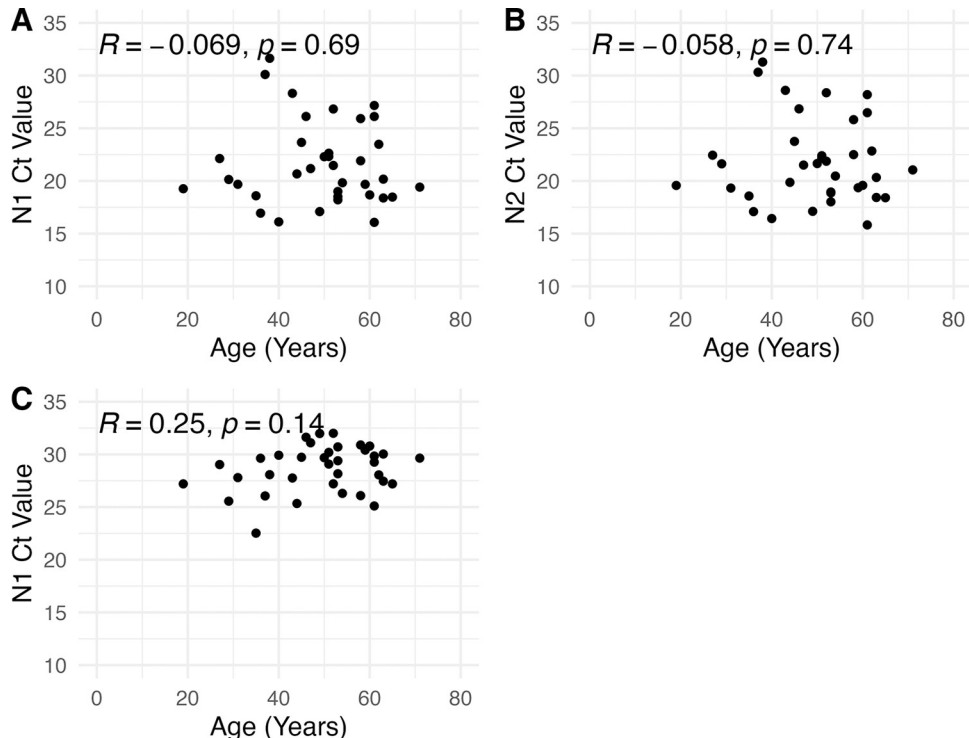

**Fig 1. Correlation between patient age and Ct values.** Age in years and Ct values for each assay were plotted and Pearson correlation determined for N1 (A), N2 (B), and Rp (C) assays.

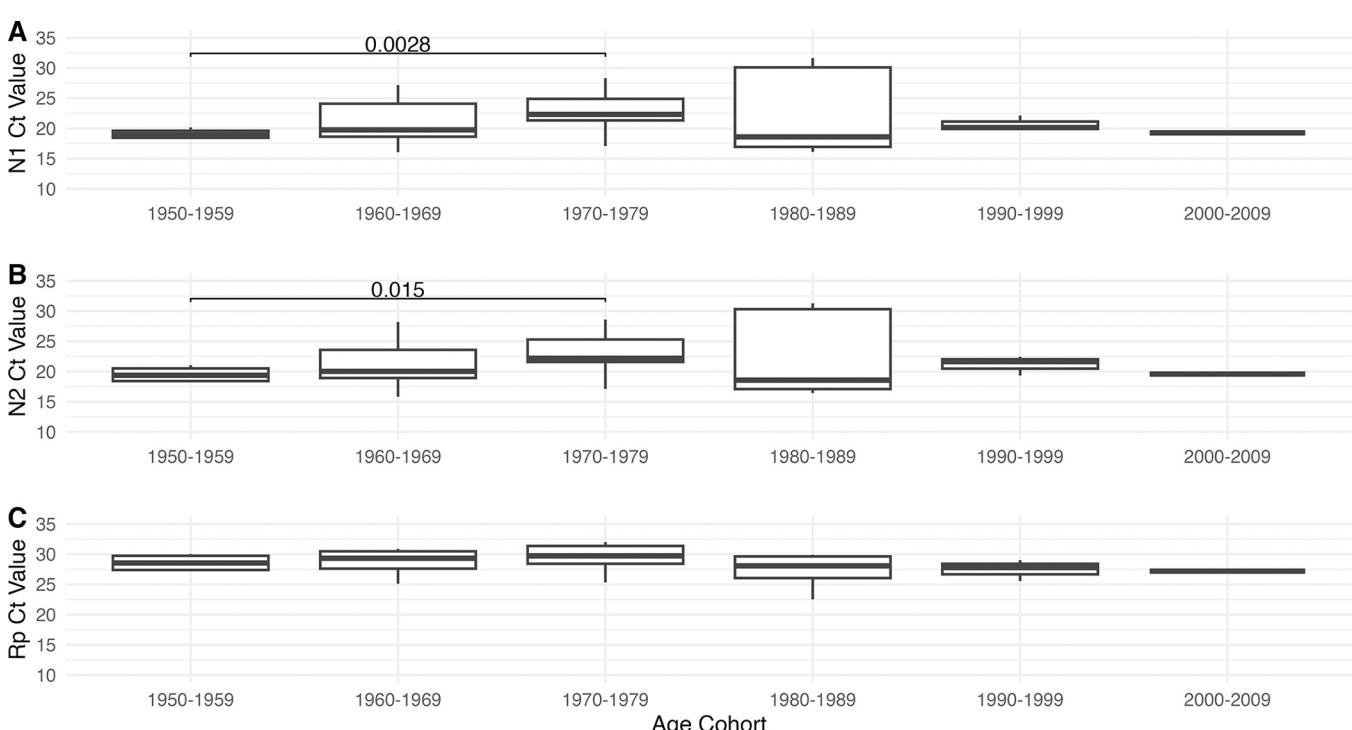

**Fig 2. Comparison of birth year by decade and Ct values.** Box and whisker plots for each decade cohort. Whiskers indicate maximum and minimum Ct value for each cohort. Horizontal line within box indicates mean Ct value. Significant ($p < 0.05$) Wilcoxon ranked-sum test significance values are indicated by brackets and listed p-values. A, N1 assay; B, N2 assay; C, Rp assay.

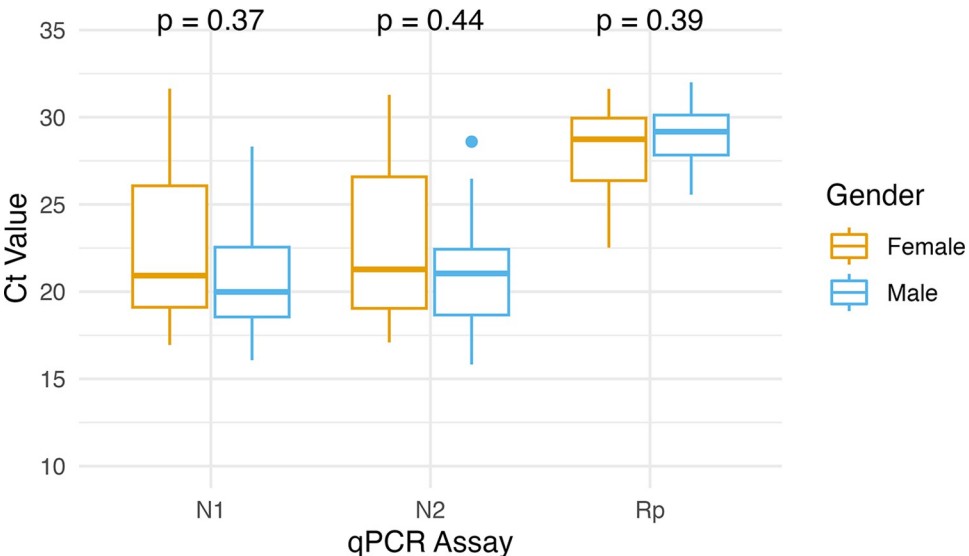

**Fig 3. Comparison of mean Ct values by gender for each assay.** Box and whisker plots for each qPCR assay by gender. Whiskers indicate maximum and minimum Ct value for each cohort. Horizontal line within box indicates mean Ct value. Dots indicate outliers. P-values of each comparison are indicated above the appropriate plot.

We also investigated whether there existed a relationship between patient gender and mean Ct values for each assay. No significant differences as determined by Wilcoxon ranked-sum testing between Ct values of male and female patients existed for any assay (Fig 3).

These results are in contrast to Levine-Tiefenbrun et al [11], who reported differences in cycle threshold for N gene detection assays between patients age 40 and above and those younger than 40 during the first four days post-diagnosis, with older patients exhibiting lower Ct values, especially in men. To investigate the possibility that a similar pattern might exist in our data, samples were sorted into the same age categories as Levine-Tiefenbrun and differences in mean Ct values were determined using Wilcoxon ranked-sum test. Interestingly, although mean Ct values were higher in the older cohort for all three assays (the opposite pattern observed by Levine-Tiefenbrun), these differences were only significant for the Rp assay (Fig 4). Removal of two samples with outlier Ct values for N1 and N2 assays, V341233359 and V341233381, did not result in statistically significant differences in these assays between age groups ($p = 0.099$ and $p = 0.109$, respectively), but did result in the difference in Ct values for Rp assay between age groups losing statistical significance ($p = 0.067$). It is not clear why the pattern of these values differs from previously published results. One possibility is a difference in variants detected within the two groups, but examination of variants within the two cohorts does not indicate this to be the case. Another potential explanation is that Levine-Tiefenbrun performed testing after a formal diagnosis of SARS-CoV-2 infection, while our study was based on patients volunteering to be tested without a formal diagnosis–which could have resulted in testing during different phases of infection. It is also not clear why the mean difference in Ct values for Rp assay is significant. Gene expression in general is known to change as a function of age in humans [e.g., 27]; we are however not aware of any studies showing age-dependent changes in expression of RNAse P itself.

To investigate the possibility of gender-specific differences within the same cohorts we further parsed Ct data for each qPCR assay into categories by male and female for a total of four cohorts: female 40 years of age or older, female younger than 40, male 40 or older, and male

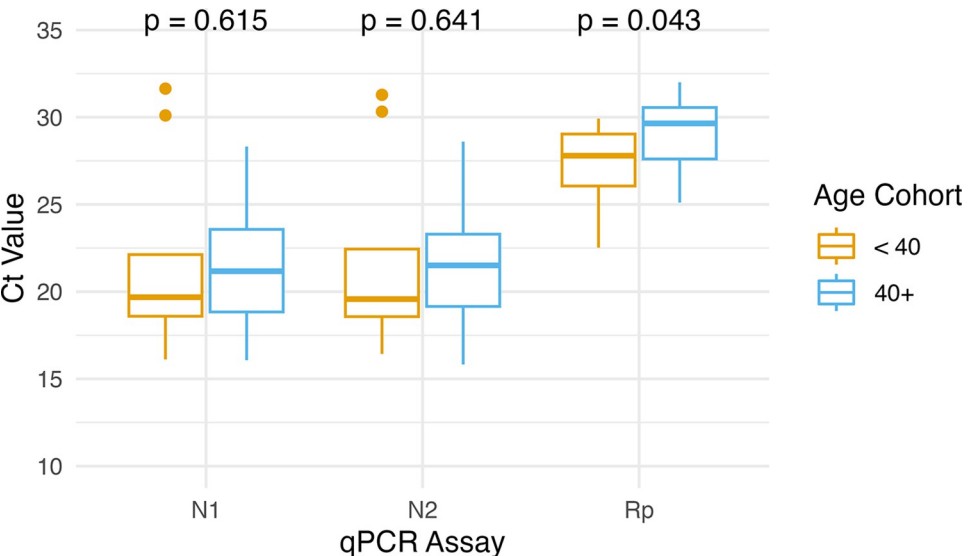

**Fig 4. Comparison of mean Ct values between samples from patients over/under 40 years of age.** Box and whisker plots showing mean Ct values for each qPCR assay, divided by age cohort as in Levine-Tiefenbrun [11]. Whiskers indicate maximum and minimum Ct value for each cohort. Horizontal line within box indicates mean Ct value. Significant ($p < 0.05$) Wilcoxon ranked-sum test significance values are listed as p-values above each assay. Dots indicate outlier samples.

younger than 40. Comparisons were made in a pairwise manner between all four cohorts. As before, although the mean Ct values for each assay were higher for the older cohort in males and females, these differences were not significant as measured by Wilcoxon ranked-sum testing. Additionally, no significantly different mean Ct values were identified between any of the six possible gender/age pairs using Wilcoxon ranked-sum testing (Fig 5). Removal of the same

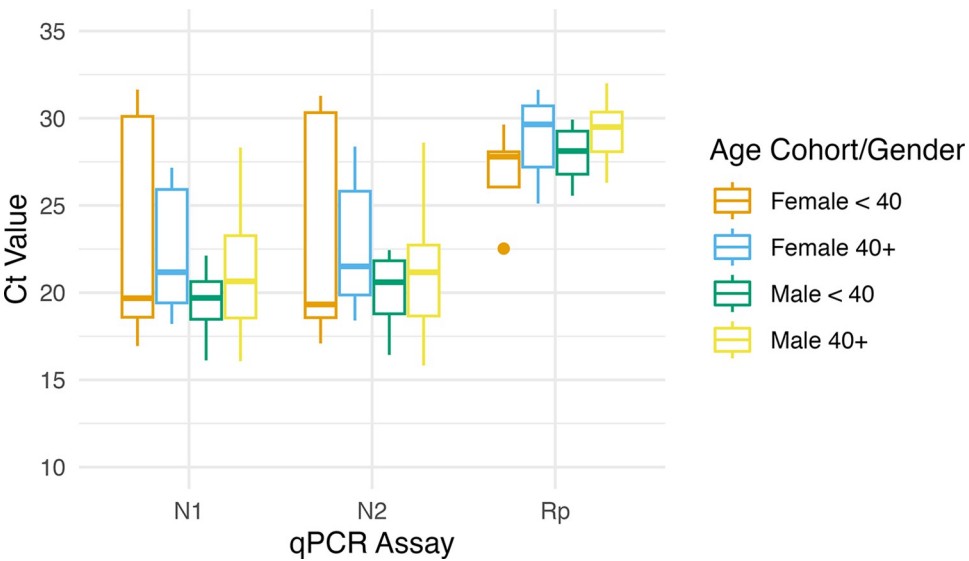

**Fig 5. Comparison of mean Ct values between age and gender cohorts.** Box and whisker plots showing mean Ct values for each qPCR assay using the same data as in Fig 3, parsed further by gender. Whiskers indicate maximum and minimum Ct value for each cohort. Horizontal line within box indicates mean Ct value. Dots indicate outlier samples.

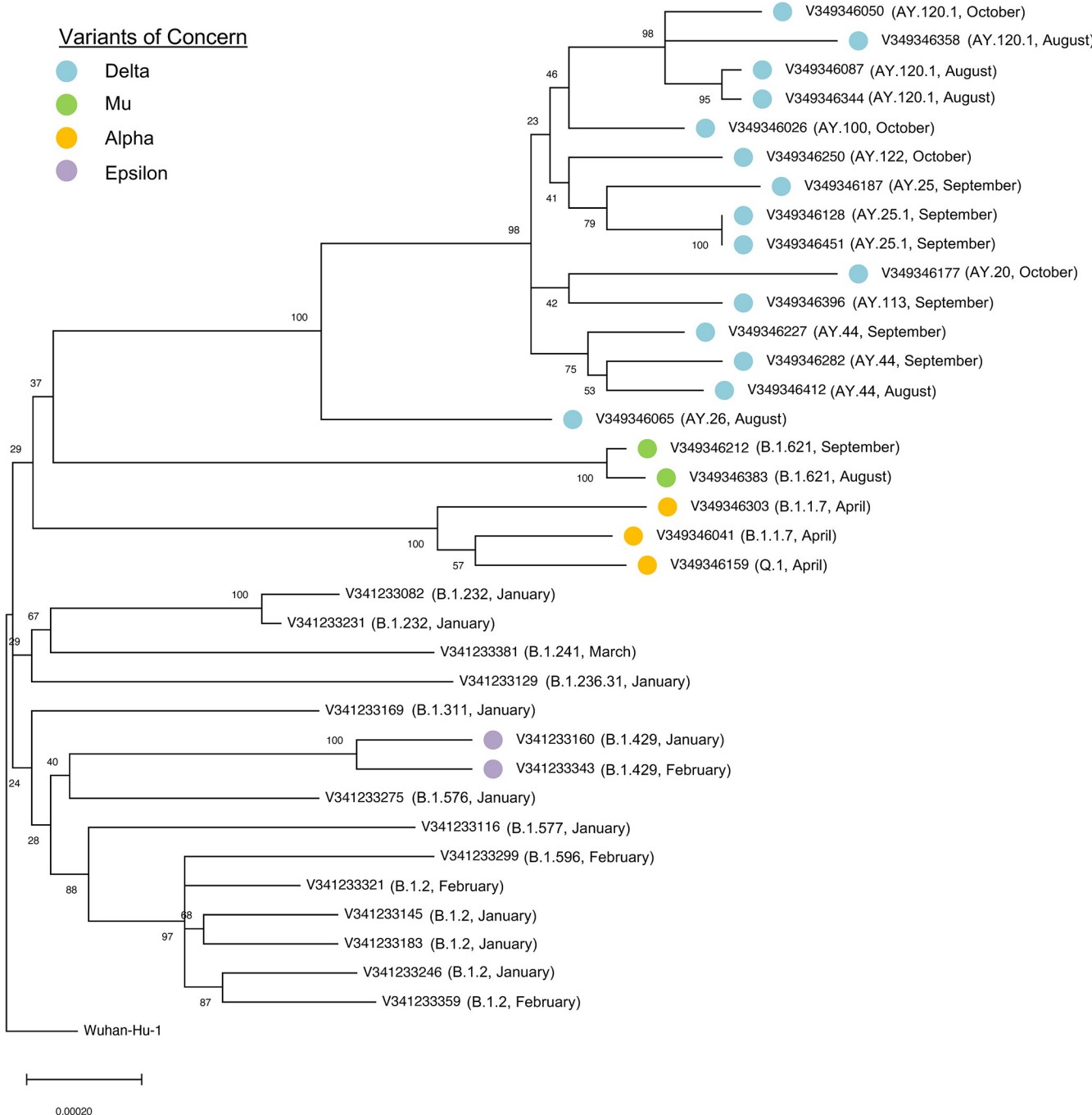

**Fig 6. Phylogeny of samples in dataset.** Maximum-likelihood tree depicting evolutionary relationship between all samples. Numbers at branch points indicate the percentage of trees in which the associated taxa clustered together during 500 bootstrap iterations. Colored circles indicate variants of concern or variants being monitored.

two outlier samples as above resulted in significant differences in N1 Ct values for females in different age groups (p = 0.025). Additionally, differences in N2 Ct values were significant for females in different age groups (p = 0.009) and for females younger than 40 compared to males 40 and over (p = 0.029). In our hands, therefore, the relationship between age or age/gender

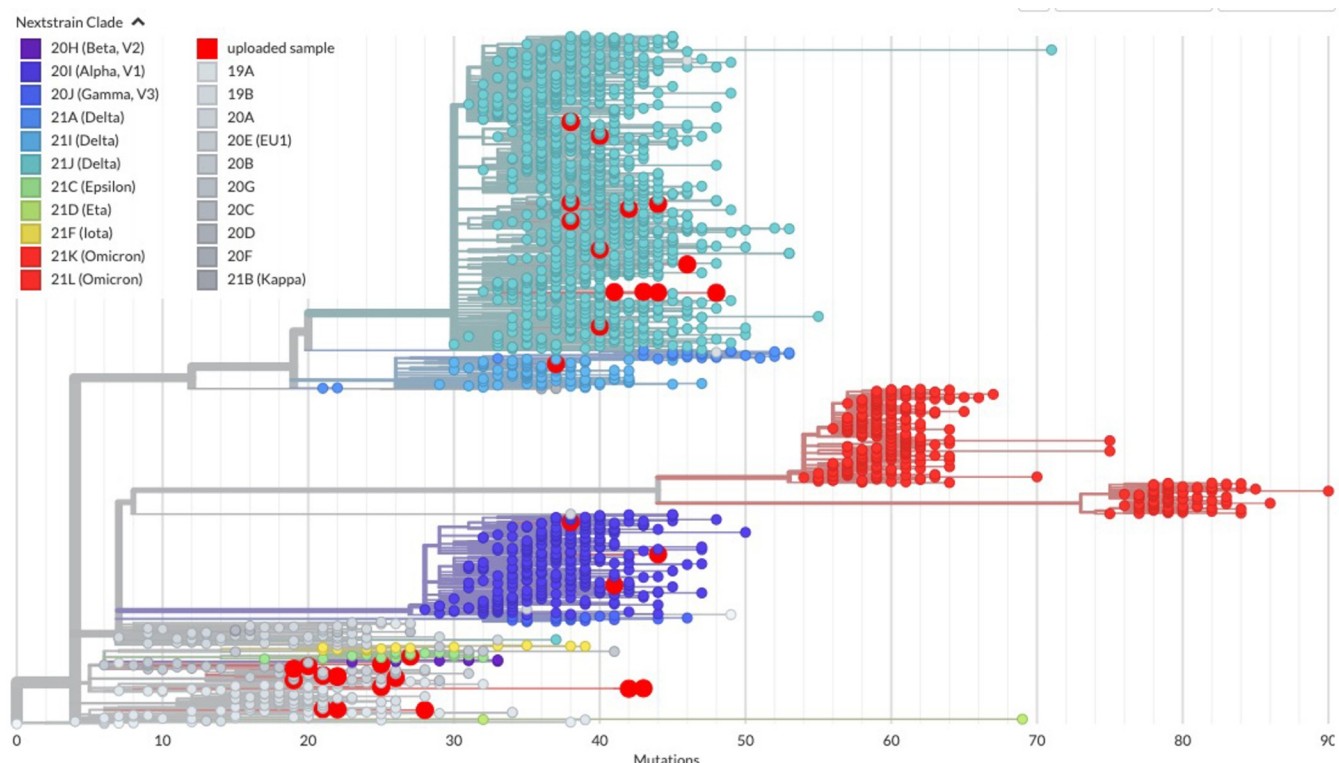

**Fig 7. Samples placed into global phylogeny.** Consensus sequences for each sample were analyzed by UShER [21] and placed into a global phylogenetic tree sorted by Nextstrain clade. Large red circles indicate uploaded samples for this dataset.

on Ct values for these assays is not strong, and it remains unclear why our results differ from those of earlier researchers.

## Analysis of SARS-COV-2 lineages

Lineages were determined by UW Medicine's Virology Laboratory, based on mutations in spike protein (Table 1). One sample, 517711, failed sequencing standards at UW and was therefore not sequenced or genotyped. Identified genotypes in this dataset correspond with circulating genotypes at the time of sampling. During early 2021, a relatively small number of samples were identified as CDC variants being monitored: two epsilon (B.1.2, V341233246; B.1.429, V341233343) and three alpha variants (B.1.1.7, V349346303; B.1.1.7, V349346041; Q.1, V349346159). However, by late 2021 two samples were identified as being a variant being monitored (V349346383 and V349346212, B.1.621/Mu) and all others were identified as a variant of concern, delta. Overall, these results indicate general concordance between this dataset and overall historical data concerning variant prevalence.

To better understand the evolutionary relationship between SARS-COV-2 samples in this dataset, a maximum-likelihood phylogeny was constructed from consensus sequences using Wuhan-Hu-1 as a root (Fig 6). Bootstrap values and low map distance in this tree indicate strong relatedness among the samples, as may be anticipated from data collected in a single geographic area. As expected, based on knowledge of variants in each sample, there was a clear break point between samples collected between January-March and August-October 2021 with three alpha variant samples emerging during April 2021. Two samples, V349346128 and V349346451 are of the same lineage (AY.25.1) and appear by this analysis to be virtually

**Table 2. Closest relatives to samples sequenced in this study.**

| Sequence ID | Closest Relative | GISAID/GenBank Accession |
|---|---|---|
| V341233082 | WA-S3600 | EPI_ISL_891026 |
| V341233116 | OR-CDC-2-3693468 | MW406623.1 |
| V341233129 | UNY-PRL-2021_0614_51O16 | EPI_ISL_2634454 |
| V341233145 | WA-UW-60057 | MW879568.1 |
| V341233160 | CA-CZB-15278 | MW564969.1 |
| V341233169 | WA-UW-48240 | EPI_ISL_824910 |
| V341233183 | OR-CDC-LC0002517 | MW635144.1 |
| V341233231 | WA-S4053 | MW555879.1 |
| V341233246 | TX-HMH-MCoV-36662 | EPI_ISL_2207284 |
| V341233275 | TG759065 | MZ910185.1 |
| V341233299 | TX-HMH-MCoV-16358 | EPI_ISL_786042 |
| V341233321 | WA-S6594 | MZ144415.1 |
| V341233343 | WA-UW-61794 | EPI_ISL_1181087 |
| V341233359 | WA-UW-61107 | EPI_ISL_1209039 |
| V341233381 | WA-UW-61900 | MZ236113.1 |
| V349346026 | WA-S14243 | OL584866.1 |
| V349346041 | TX-TCH-TCMC03123 | EPI_ISL_1621264 |
| V349346050 | WA-CDC-UW21100203750 | OK534456.1 |
| V349346065 | WA-CDC-UW21080316314 | MZ882533.1 |
| V349346087 | WA-CDC-LC0213643 | OK116997.1 |
| V349346128 | WA-CDC-UW21090701835 | OK301508.1 |
| V349346159 | PA-CDC-LC0020518 | MW783536.1 |
| V349346177 | WA-CDC-UW21101098858 | OK640401.1 |
| V349346187 | VA-CAV_VAS3N_00004268_01 | EPI_ISL_6865654 |
| V349346212 | WA-CDC-LC0281523 | OK368114.1 |
| V349346227 | WA-CDC-UW21091008798 | OK399889.1 |
| V349346250 | WA-CDC-UW21100124065 | OK534300.1 |
| V349346282 | WA-CDC-LC0181680 | MZ993119.1 |
| V349346303 | CA-CDC-STM-000042142 | OK192961.1 |
| V349346344 | WA-CDC-LC0213643 | OK116997.1 |
| V349346358 | WA-UW-21071353973 | EPI_ISL_3161738 |
| V349346383 | WA-CDC-LC0281523 | OK368114.1 |
| V349346396 | MI-MDHHS-SC34651 | OL983723.1 |
| V349346412 | WA-CDC-UW21080101922 | MZ882755.1 |
| V349346451 | WA-CDC-UW21090701835 | OK301508.1 |

identical, raising the possibility of transmission between close contacts; during this time period, one work-based case of transmission was documented. An additional sample, V349346065 has a strong bootstrap value placing it outside the main delta variant clade, suggesting the possibility that this infection took place outside of the immediate geographic area.

Finally, consensus sequences were placed into an existing global phylogeny comprising 8,790,585 worldwide genomes using UShER [21] (Fig 7). In this analysis, samples fall into the expected clades by variant (e.g., delta clade, alpha clade, etc.). Interestingly, samples from this effort in many cases did not cluster together but were in fact placed into divergent lineages within clades. To investigate this phenomenon more fully, closest genetic relatives were identified (Table 2). In many cases the closest relative to a given sample as determined by UShER were not from Washington state, indicating infection in other regions within the United States

with subsequent testing in WA, or possibly infection within WA from sources traveling from elsewhere. Geographic separation during infection would also explain why the samples shown in Fig 7 do not cluster together.

Taken together, the results presented here provide a snapshot into regional SARS-COV-2 transmission within Washington state, USA during the spring and fall of 2021. While there was no overall correlation between patient age and viral load as measured by qPCR Ct values, certain cohorts of patient samples did exhibit subtly different Ct values for N1 and N2 assays. However, the significance of these differences is generally low and similar to differences in Rp control gene expression between the same cohorts. Additionally, no significant relationship was found between patient age and sampling date or variant identified, between sampling date and Ct values, or between Ct values and patient gender. Additional insight into transmission dynamics during this period was obtained by placing each sample into a global phylogenetic context, which indicated significant geographic variability. Consistent with observed geographic variability, there was only a single confirmed case of workplace transmission in the samples examined, and together these highlight the risks posed by travel during COVID-19 surges.

## Acknowledgments

The authors wish to thank Pavitra Roychoudhury for helpful discussions of data availability and global SARS-COV-2 genome repositories.

## Author Contributions

**Conceptualization:** Kristin M. Omberg.

**Formal analysis:** Owen P. Leiser.

**Investigation:** Deanna L. Auberry, Erica Bakker, Will Chrisler, Kristin Engbrecht, Heather Engelmann, Sarah Fansler, Vincent Gerbasi, Joshua Hansen, Chelsea Hutchinson, Janine Hutchison, Mary J. Lancaster, Kathleen Lawrence, Angela Melville, Jennifer Mobberley, Isabelle O'Bryon, Kristie L. Oxford, Tessa Oxford, Shelby Phillips, Kabrena E. Rodda, James A. Sanford, Athena Schepmoes, Brian E. Staley, Kelcey Terrell, Kristin Victry, Cynthia Warner.

**Supervision:** Kristin M. Omberg.

**Visualization:** Owen P. Leiser.

**Writing – original draft:** Owen P. Leiser.

**Writing – review & editing:** Owen P. Leiser, Deanna L. Auberry, Erica Bakker, Will Chrisler, Kristin Engbrecht, Heather Engelmann, Sarah Fansler, Vincent Gerbasi, Joshua Hansen, Chelsea Hutchinson, Janine Hutchison, Mary J. Lancaster, Kathleen Lawrence, Angela Melville, Jennifer Mobberley, Isabelle O'Bryon, Kristie L. Oxford, Tessa Oxford, Shelby Phillips, Kabrena E. Rodda, James A. Sanford, Athena Schepmoes, Brian E. Staley, Kelcey Terrell, Kristin Victry, Cynthia Warner, Kristin M. Omberg.

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
