## [Decision Letter · Decision Letter 0]

13 Feb 2023

PONE-D-23-00183Insights from a workplace SARS-CoV-2 specimen collection program, with genomes placed into global sequence phylogenyPLOS ONE

Dear Dr. Omberg,

Thank you for submitting your manuscript to PLOS ONE. After careful consideration, we feel that it has merit but does not fully meet PLOS ONE’s publication criteria as it currently stands. Therefore, we invite you to submit a revised version of the manuscript that addresses the points raised during the review process.

We look forward to receiving your revised manuscript.

Kind regards,

Babatunde Olanrewaju Motayo, Ph.D.

Academic Editor

PLOS ONE

Journal Requirements:

2. Please make sure that all information entered in the 'Ethics Statement' section regarding ethics approval and informed participant consent is also included in the Methods section of the manuscript.

Additional Editor Comments:

I have received the review reports on your manuscript titled "Insights from a workplace SARS-CoV-2 specimen collection program, with genomes placed into global sequence phylogeny" which you submitted to PLOSONE.

Based on the advice received, your manuscript could be accepted for publication should you be prepared to incorporate the suggested minor revisions.

When preparing your revised manuscript, you are to carefully consider all of the reviewer comments, and submit a point-by-point list of responses to the comments.

Your list of responses should be uploaded as a file in addition to your revised manuscript.

Important: ALL CHANGES MADE TO THE MANUSCRIPT MUST BE IN "TRACK CHANGE", RED FONT OR HIGHLIGHTED FONT.

Failure to do so will result in rejection of this manuscript.

Also, you are requested to submit both the clean version of the revised manuscript and the tracked version of the same.

In order to submit your revised manuscript electronically, please access the Editorial Manager Website.

Reviewers' comments:

Reviewer's Responses to Questions

**Comments to the Author**

1. Is the manuscript technically sound, and do the data support the conclusions?

Reviewer #1: Yes

2. Has the statistical analysis been performed appropriately and rigorously? 

Reviewer #1: Yes

3. Have the authors made all data underlying the findings in their manuscript fully available?

Reviewer #1: Yes

4. Is the manuscript presented in an intelligible fashion and written in standard English?

Reviewer #1: Yes

5. Review Comments to the Author

Reviewer #1: The research has added to knowledge about SARS-CoV-2 infection and in concept of susceptility generally.

1. The Abstract should be more defined by mentioning the pool by percentage where the 36 positives emerged.

2. The positive samples tested and analyzed should have a prevalence or percentage from the population tested and or the samples are selected a pool of positives which should be mentioned based on Ct values ranges or symptoms manifested by subjects.

3. There is no need to indent the paragraph, spacing adequately is ideal.

Line 71- Viral RNA extracted (NOT Isolated).

Line 34 – Statistical (NOT Statical)

4. The statistical analysis was properly done but some of the figure representation are not properly presented especially figure 5 should re-presented using another template to represent the Age Cohort / Gender clearly.

The figures should be labelled underneath of each of the figures with a written legend.

6. PLOS authors have the option to publish the peer review history of their article (what does this mean?). If published, this will include your full peer review and any attached files.

Reviewer #1: **Yes: **Bamidele Soji Oderinde, Ph.D

---

## [Author Response · Author response to Decision Letter 0]

27 Mar 2023

Reviewer #1: The research has added to knowledge about SARS-CoV-2 infection and in concept of susceptibility generally.

1. The Abstract should be more defined by mentioning the pool by percentage where the 36 positives emerged.

Response: We have edited the abstract to include this information.

2. The positive samples tested and analyzed should have a prevalence or percentage from the population tested and or the samples are selected a pool of positives which should be mentioned based on Ct values ranges or symptoms manifested by subjects.

Response: We have included information regarding percent positive of total screened patients. Given the voluntary nature of testing for this study, it would be reasonable to assume that patients experienced symptoms consistent with flu-like illness. However, because these symptoms were not recorded for this dataset, we have not made references to clinical symptoms in the manuscript.

3. There is no need to indent the paragraph, spacing adequately is ideal.

Response: We have left paragraphs indented in the body of the manuscript in accordance with journal style guidelines.

Line 71- Viral RNA extracted (NOT Isolated).

Line 34 – Statistical (NOT Statical)

Response: We are unsure exactly to which lines the reviewer refers but have made changes to isolated/extracted at line 97 and have corrected the statical/statistical typo at line 111.

4. The statistical analysis was properly done but some of the figure representations are not properly presented especially figure 5 should re-presented using another template to represent the Age Cohort / Gender clearly.

The figures should be labelled underneath of each of the figures with a written legend.

Response: We thank the reviewer for the comment regarding figure presentation, but respectfully disagree that the representation of age/gender cohort is not presented clearly in this figure. This figure was designed such that multiple comparisons between age/gender cohort can be made visually by the reader from a single figure – other representations of the data such as scatter plots do not reflect the distribution of and mean Ct values for each assay while representations such as violin plots do not allow for quick visual comparison between cohort pairs.

Each cohort of age and gender is clearly notated by distinct colors, chosen from a palate specifically designed to be distinguishable by readers with color-blindness. Additionally, the figure legend explicitly describes horizontal lines as mean Ct values for each qPCR assay, outliers, etc. However, we have adjusted the text in this section to define cohorts being tested more clearly as well as their relationships to each other. Finally, the figures are not labeled as submitted because PLOS ONE style guidelines are specific that figure legends should appear in the body of the manuscript after the paragraph in which each figure is referenced.

---

## [Decision Letter · Decision Letter 1]

10 Apr 2023

PONE-D-23-00183R1Insights from a workplace SARS-CoV-2 specimen collection program, with genomes placed into global sequence phylogenyPLOS ONE

Dear Dr. Omberg,

Thank you for submitting your manuscript to PLOS ONE. After careful consideration, we feel that it has merit but does not fully meet PLOS ONE’s publication criteria as it currently stands. Therefore, we invite you to submit a revised version of the manuscript that addresses the points raised during the review process.

We look forward to receiving your revised manuscript.

Kind regards,

Babatunde Olanrewaju Motayo, Ph.D.

Academic Editor

PLOS ONE

Journal Requirements:

Reviewers' comments:

Reviewer's Responses to Questions

**Comments to the Author**

1. If the authors have adequately addressed your comments raised in a previous round of review and you feel that this manuscript is now acceptable for publication, you may indicate that here to bypass the “Comments to the Author” section, enter your conflict of interest statement in the “Confidential to Editor” section, and submit your "Accept" recommendation.

Reviewer #1: (No Response)

2. Is the manuscript technically sound, and do the data support the conclusions?

Reviewer #1: Yes

3. Has the statistical analysis been performed appropriately and rigorously? 

Reviewer #1: Yes

4. Have the authors made all data underlying the findings in their manuscript fully available?

Reviewer #1: Yes

5. Is the manuscript presented in an intelligible fashion and written in standard English?

Reviewer #1: Yes

6. Review Comments to the Author

Reviewer #1: the review for the author is to address the samples tested in a scientific way:

The 36 out of the 64 positive samples tested explanation should be more scientific. That is the 36 tested should either be selected based on ct value range or the severity of symptoms of the patients whose samples were tested. But if all sample ct values are of same range then there has to be random selection of the sample.

Once that is addressed, the article can be published.

7. PLOS authors have the option to publish the peer review history of their article (what does this mean?). If published, this will include your full peer review and any attached files.

Reviewer #1: **Yes: **Dr B.S. Oderinde

---

## [Author Response · Author response to Decision Letter 1]

12 Apr 2023

6. Review Comments to the Author

Reviewer #1: the review for the author is to address the samples tested in a scientific way:

The 36 out of the 64 positive samples tested explanation should be more scientific. That is the 36 tested should either be selected based on ct value range or the severity of symptoms of the patients whose samples were tested. But if all sample ct values are of same range then there has to be random selection of the sample.

Once that is addressed, the article can be published.

Response: We have edited the abstract and the methods to state that all 36 positive samples released for research were sequenced and genotyped. We hope this helps to clarify that 64 total samples were positive; but of those 64, only 36 could be sequenced under the terms of our IRB approval.

---

## [Editor Report · Decision Letter 2]

14 Apr 2023

Insights from a workplace SARS-CoV-2 specimen collection program, with genomes placed into global sequence phylogeny

PONE-D-23-00183R2

Dear Dr. Omberg,

We’re pleased to inform you that your manuscript has been judged scientifically suitable for publication and will be formally accepted for publication once it meets all outstanding technical requirements.

Kind regards,

Babatunde Olanrewaju Motayo, Ph.D.

Academic Editor

PLOS ONE
---

## [Editor Report · Acceptance letter]

20 Apr 2023

PONE-D-23-00183R2 

Insights from a workplace SARS-CoV-2 specimen collection program, with genomes placed into global sequence phylogeny 

Dear Dr. Omberg:

I'm pleased to inform you that your manuscript has been deemed suitable for publication in PLOS ONE. Congratulations! Your manuscript is now with our production department. 

Kind regards, 

on behalf of

Dr Babatunde Olanrewaju Motayo 

Academic Editor

PLOS ONE